# Modulation of Allicin-Free Garlic on Gut Microbiome

**DOI:** 10.3390/molecules25030682

**Published:** 2020-02-05

**Authors:** Keyu Chen, Yasushi Nakasone, Kun Xie, Kozue Sakao, De-Xing Hou

**Affiliations:** 1The United Graduate School of Agricultural Sciences, Kagoshima University, Kagoshima 890-0065, Japan; k4164345@kadai.JP (K.C.); k3591458@kadai.jp (K.X.); sakaok24@chem.agri.kagoshima-u.ac.jp (K.S.); 2Kenkoukazoku Co., Kagoshima 892-0848, Japan; ymine@uoguelph.ca; 3The United Graduate School of Agriculture, Forest and Fishery Science, Faculty of Agriculture, Kagoshima University, Kagoshima 890-0065, Japan

**Keywords:** garlic, allicin-free, gut microbiome

## Abstract

The allicin diallyldisulfid-*S*-oxide, a major garlic organosulfur compound (OSC) in crushed garlic (*Allium sativum* L.), possesses antibacterial effects, and influences gut bacteria. In this study, we made allicin-free garlic (AFG) extract and investigated its effects on gut microbiome. C57BL/6N male mice were randomly divided into 6 groups and fed normal diet (ND) and high-fat diet (HFD) supplemented with or without AFG in concentrations of 1% and 5% for 11 weeks. The genomic DNAs of feces were used to identify the gut microbiome by sequencing 16S rRNA genes. The results revealed that the ratio of p-*Firmicutes* to p-*Bacteroidetes* increased by aging and HFD was reduced by AFG. In particular, the f-*Lachnospiraceae*, g-*Akkermansia*, and g-*Lactobacillus* decreased by aging and HFD was enhanced by AFG. The g-*Dorea* increased by aging and HFD decreased by AFG. In addition, the ratio of glutamic-pyruvic transaminase to glutamic-oxaloacetic transaminase (GPT/GOT) in serum was significantly increased in the HFD group and decreased by AFG. In summary, our data demonstrated that dietary intervention with AFG is a potential way to balance the gut microbiome disturbed by a high-fat diet.

## 1. Introduction

Gut microbiome plays an important role in maintaining a healthy body [1]. Diet has been shown to impact the composition and activity of gut microbiota [2]. A high-fat diet (HFD) modulates the gut microbiome composition by decreasing the prevalence of specific gut barrier-protecting bacteria and increasing the prevalence of opportunistic pathogens that can release free antigens such as lipopolysaccharides. This imbalance may be associated with higher gut permeability, leading to higher plasma levels of endotoxin and inflammation factors, and eventual development of metabolic disorders [3,4]. Simultaneously, fluctuation of gut microbiome was also affected by aging. For example, the ratio of *Firmicutes*/*Bacteroidetes* and the abundance of g-*Akkermansia*, which is considered to maintain the integrity of the intestinal barrier, were changed along with aging [5,6,7].

Garlic (*Allium sativum* L.) has long been used in food and medicine. Most of the carbohydrates in garlic are composed of a water-soluble fructose polymer called fructan [8], accounting for approximately 65% of the dry weight [9]. On the other hand, garlic contains 1.1–3.5% organosulfur compounds (OSCs), which is far higher than that in other plant food. The primary OSCs are γ-glutamyl-*S*-allyl-*L*-cysteines (GSAC), which are hydrolyzed and oxidized to yield *S*-allyl-*L*-cysteine sulfoxides (alliin) during storage [10]. The alliin is a major *S*-alk(en)ylcysteine sulfoxide compound which is stored in the mesophyll cells of garlic. When garlic bulbs are crushed, cut, or ground, alliinase is released from the bundle sheath cells to catalyze alliin into a reactive intermediate, sulfenic acid, pyruvic acid, and ammonia [11]. Sulfenic acid undergoes self-condensation to produce allicin, which is then decomposed into other many OSCs, such as diallyl sulfide series, thiosulfates, and ajoene (Appendix A) [12].

The experiments with separated garlic compounds revealed that fructans work as prebiotics for gut microbiome [13], while garlic OSCs, such as allicin, thiosulfates, and ajoene, act as antibacterial effects [14,15]. The complicated ingredients of garlic seem to give paradoxical results for the gut microbiome. To develop garlic supplements that are beneficial to the gut microbiome, we made allicin-free garlic (AFG) extract by heating a garlic bulb at 80 °C to inactivate alliinase for blocking allicin production, and then investigated its effect on the gut microbiome in a mouse model fed with normal diet (ND) or HFD.

## 2. Result

### 2.1. Body Weight and Index of Liver Injury

There was no difference in initial body weight and in daily food intake among the groups. The final body weight of mice fed with HFD was significantly higher than the mice fed with ND (ND: 35.9 ± 3.34 g, HFD: 43.2 ± 0.67 g, *p* < 0.05). AFG supplementation had no significant effect on body weight in both ND and HFD groups (Figure 1A). Moreover, the ratio of GPT to GOT in serum increased significantly (*p* < 0.05) in HFD group and reduced significantly (*p* < 0.05) by AFG supplementation (Figure 1B). These data indicated that the dose of garlic used in this experiment has no damage to liver and might attenuate the HFD-induced burden of liver since GPT and GOT enzymes usually leak out into the general circulation when liver cells are injured.

### 2.2. Effect of AFG on Lipid Metabolism

To clarify the effect of AFG on lipid metabolism, we measured the serum levels of lipid and glucose metabolism markers at the final day of the experiment after 12 h fasting. As shown in Figure 2A–D, the serum levels of triacylglycerol (TG), total cholesterol (T-Cho), and low-density lipoproteins (LDL) increased significantly in HFD group (*p* < 0.05), and decreased slightly by AFG although there was no significant difference.

### 2.3. Effect of AFG on the Diversity of Gut Microbiome

To further understand the effects of AFG on gut bacteria, the composition and relative abundance of microbiota were determined, using high-throughput 16S rRNA gene sequencing. The taxa richness of the gut microbiome was assessed by α-diversity analyses using Chao1 value, observed species index, PD whole tree index, and Shannon index. As shown in Figure 3A–D, the α-diversity of gut microbiome increased by aging and HFD diet, and restored by AFG supplementation. Moreover, we used principle coordinate analyses (PCoA) plot (β-diversity: between-habitat diversity) based on unweighted UniFrac distance matrices to investigate the similarities in gut microbial community structure among different groups. Percent of dataset variability explained by each principal coordinate is shown in the axis’s titles (PC1:11.24%, PC2:9.95%, PC3:7.32%). The PCoA plot indicated that the structure of gut microbiota in the ND group was changed along with aging (Figure 3E–F).

### 2.4. Modulation of Gut Microbiome by AFG

To know the modulation of AFG on gut microbiome, first we investigated the changes of individual microbial species at the phylum level. The phylum of *Verrucomicrobia* decreased by aging and HFD diet, and recovered by 5% concentration of AFG supplementation (Figure 4A). Moreover, the ratio of *p-Firmicutes* to *p-Bacteroidetes* in the ND group increased with aging from 7 weeks to 18 weeks, and decreased by 5% AFG supplementation. This ratio was also increased by HFD, and reduced by 5% AFG supplementation (Figure 4B).

Secondly, we analyzed the changes of individual microbial species at genus level (Figure 5A). The results revealed that the relative abundance of g-*Akkermansia* belonging to the phylum of *Verrucomicrobia* decreased by aging and HFD diet, and recovered by AFG supplementation. Furthermore, the relative abundance of f-*Lachnospiraceae* was also decreased by aging and HFD diet, and increased by AFG supplementation.

## 3. Discussion

This study revealed the preventive effects of AFG supplementation on the HFD-induced hepatocyte damage and the dysbiosis of the gut microbiome. Those effects were that AFG attenuated HFD-induced increases in the ratio of GPT/GOT and the ratio of p-*Firmicutes* to p-*Bacteroidetes*. The dysbiosis of gut microbiota is found in a lot of diseases; Non-alcoholic fatty liver disease (NAFLD) is one of them in humans. Thus, targeting microbiota and their metabolites have recently been developed to address this issue [16]. Obese people are found to have a higher ratio of *Firmicutes*/*Bacteroidetes*, compared to normal-weight people in the adult population [17]. Our data revealed that HFD diet increased the ratio of *Firmicutes*/*Bacteroidetes*, and this increase was significantly inhibited by 5% AFG.

Furthermore, the relative abundance of f-*Lachnospiraceae* was decreased by HFD, but increased by AFG. f-*Lachnospiraceae* has been reported to be associated with anti-inflammatory activity [18], host mucosal integrity [19], the consumption of energy and the level of leptin [20]. Fructan [21,22] and whole garlic [23] have been reported to increase the abundance of f-*Lachnospiraceae*, but alliin was found to decreased it [24]. In particular, the relative abundance of g-*Akkermansia* decreased by aging and HFD diet, and increased by 5% AFG supplementation. g-*Akkermansia* is associated with the reduction of gut leakiness and attenuation of low-grade inflammation [25], and considered to be next-generation beneficial bacteria [26,27]. Moreover, the relative abundance of f-*Lactobacillus* was enhanced by 5% AFG, and it has been reported that oral *Lactobacillus* tablets decreased the levels of GOT and GPT in patients with NAFLD in two double-blind randomized clinical trials [28,29]. Interestingly, the growth of f-*Lactobacillus* was mostly unaffected by the addition of raw garlic containing allicin compared with other gastrointestinal symbiotic bacteria in vitro culture experiments [30]; however, it was enhanced by AFG in this study. Thus, it may be due to fructan effects, a major beneficial ingredient for gut bacterial growth.

To understand the effects of allicin on the gut microbiome, we compared the whole garlic extract from our previous experiments [23]. We found that whole garlic and AFG had the same tendency to inhibit the ratio of GPT to GOT caused by HFD. Although we found that whole garlic inhibited the HFD-induced increases of TG and LDL [23], AFG had no significant effect on lipid metabolism in this study. These data suggested that the garlic OSCs played an important role in improvement of HFD-induced dyslipidemia. On the other hand, although both of the whole garlic [23] and AFG increased the relative abundance of f-*Lachnospiraceae*, g-*Akkermansia*, and g-*Lactobacillus*, AFG showed down-regulation of the diversity of gut bacteria, and whole garlic [23] showed up-regulation of the diversity of gut bacteria. It is possible that this is due to the interaction effect of garlic OSCs and fructans on gut microbiome, although the reason is still unknown.

In summary, the supplementation with 1–5% of AFG in diet significantly decreased the ratio of GPT/GOT induced by HFD in serum. Fecal microbiota characterization by high-throughput 16S rRNA gene sequencing demonstrated that AFG reduced the *Firmicutes/Bacteroidetes* ratio caused by aging and the ingestion of HFD. Moreover, the abundance of f-*Lachnospiraceae*, g-*Akkermansia,* and g-*Lactobacillus* were enhanced by AFG. Our data demonstrated that allicin-free garlic is better than whole garlic for the modulation of the gut microbiome.

## 4. Materials and Methods

### 4.1. Chemicals and Reagents

Garlic was harvested from Aomori Prefecture, Japan. To inactivate alliinase and prevent the formation of allicin, raw garlic was heated at 80 °C for 1 h in a water ratio of 1:5. After removing water, heated garlic was homogenized and then centrifuged 4 times at 5000 rpm for 30 min. The supernatant was collected and freeze-dried to garlic powder, which was further washed with ethanol and dried again at 60 °C for 40 min to obtain AFG powder. The recovery rate was 14%. The amount of OSCs and fructans on garlic powder was determined by high performance liquid chromatography (HPLC) or fructans assay kit (Biocon Ltd., Nagoya, Japan), respectively (Appendix A).

The nutrient composition of the diets is shown in Appendix A. All ND contained 21% protein, 6% fat and 54% carbohydrate, 4% cellulose and about 370 kcal/100 g total calories. All HFD contained 21% protein, 40% fat and 10% carbohydrate, 4% cellulose and about 570 kcal/100 g total calories. Lard oil was obtained from Sigma–Aldrich Japan (Tokyo, Japan).

### 4.2. Mouse Model

The animal experimental protocol was drafted according to the guidelines of the Animal Care and Use Committee of Kagoshima University (Permission NO. A12005). Male C57 BL/6N mice (5 weeks of age) from Japan SLC Inc. (Shizuoka, Japan) were housed separately in cages with wood s havings bedding under controlled light (12-h light/day) and temperature (25 °C), where they had free access to water and feed. Mice body weight was weighed once a week. After 14 day of acclimatization (7 weeks of age), the mice were randomly divided into six groups (*n* = 4) and fed with ND, ND + AFG (allicin-free garlic supplement) with different concentration of 1% or 5%, HFD (40% of fat), HFD + AFG with different concentration of 1% or 5% (Appendix A). After 11 weeks of feeding (18 weeks of age), mice were sacrificed after overnight fasting. The fresh feces were collected at the beginning (7 weeks of age) and the end of the experiment (18 weeks of age).

### 4.3. Measurement of Serum Biochemical Indicators

Blood was obtained from mice eyeballs and collected in the tube with a coagulant (Separable microtubes, FUCHIGAMI,170720, Kyoto, Japan) for 30 min at room temperature to coagulate properly and acquired by centrifuging at 3000 rpm for 5 min and stored at −80 °C until use. The serum levels of glutamic-oxaloacetic transaminase (GOT), glutamic-pyruvic transaminase (GPT), gamma-glutamyl transferase (GGT), total cholesterol (T-Cho), total triacylglycerol (TG), high-density lipoprotein cholesterol (HDL-c) and glucose were measured with an automated analyzer for clinical chemistry (SPOTCHEM EZ SP-4430, Arkray, Kyoto, Japan). Using the Friedewald equation that LDL = Tcho-HDLc-TG/5 to calculate the level of LDL [31].

### 4.4. Characterization of Gut Microbiome by 16S rRNA Gene Sequencing

Mice feces were collected from mice housed in different cages at 7- and 18-weeks age, and soon stored at −80 °C until use. The feces genomic DNA was extracted with the Fast DNA spin kit (MP BIOMEDICALS, Kyoto, Japan) according to the manufacturer’s manual, for analyzing the composition of gut bacterial communities by sequencing 16S rRNA genes as described in our previous paper [32].

### 4.5. Statistical Analysis

Results were expressed as mean ± SD. The significant differences between the groups were analyzed by one-way analysis of variance (ANOVA) tests, followed by Duncan’s multiple range tests with the SPSS statistical program (version 19.0, IBM Corp., Armonk, NY, USA). A probability of *p* < 0.05 was considered significant.

## Figures and Tables

**Figure 1 molecules-25-00682-f001:**
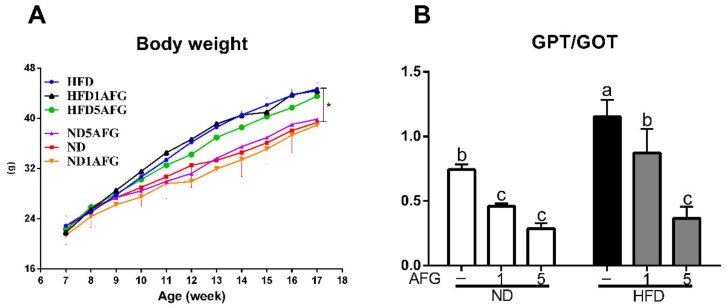
Body weight and index of liver injury. (**A**) The changes in body weight of mice fed with different diets from 7 to 18 weeks. Results are expressed as the mean ± SD for each group of mice (n = 4). The asterisk indicates a significant difference in weight at 18 weeks (*p* < 0.05). HFD: high-fat diet, HFD1AFG: HFD plus 1% AFG, HFD5AFG: HFD plus 5% AFG, ND: normal diet, ND1AFG: ND plus 1% AFG, ND5AFG: ND plus 5% AFG. (**B**) The ratio of serum GPT/GOT. The data represent the mean ± SD of four mice. The different letter indicates a significant difference at *p* < 0.05.

**Figure 2 molecules-25-00682-f002:**
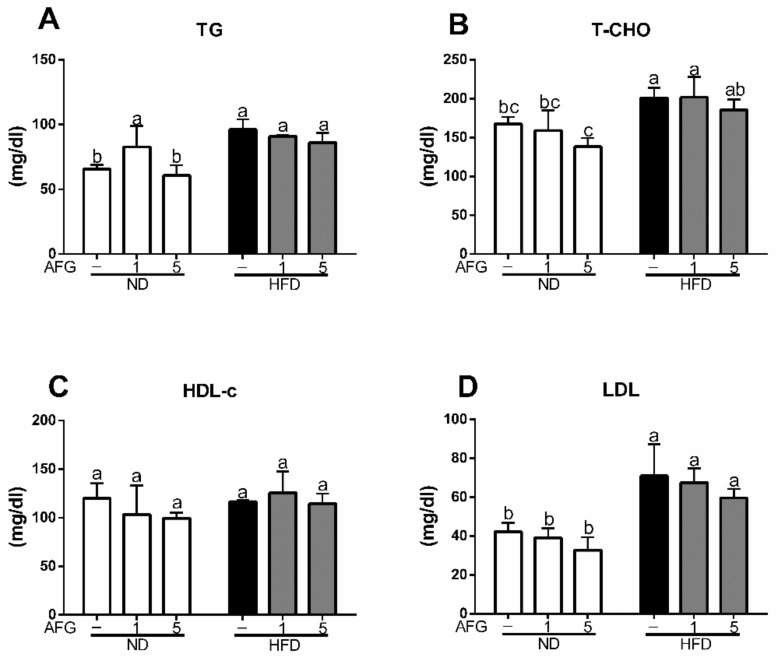
Effect of allicin-free garlic on lipid metabolism. (**A**–**D**) Influence of allicin-free garlic supplementation on serum level of total triacylglycerol (TG), total cholesterol (T-Cho), high-density lipoprotein cholesterol (HDL-c) and low-density lipoproteins (LDL). The data represent the mean ± SD of four mice for each group. Columns with different letters significantly differ (*p* < 0.05).

**Figure 3 molecules-25-00682-f003:**
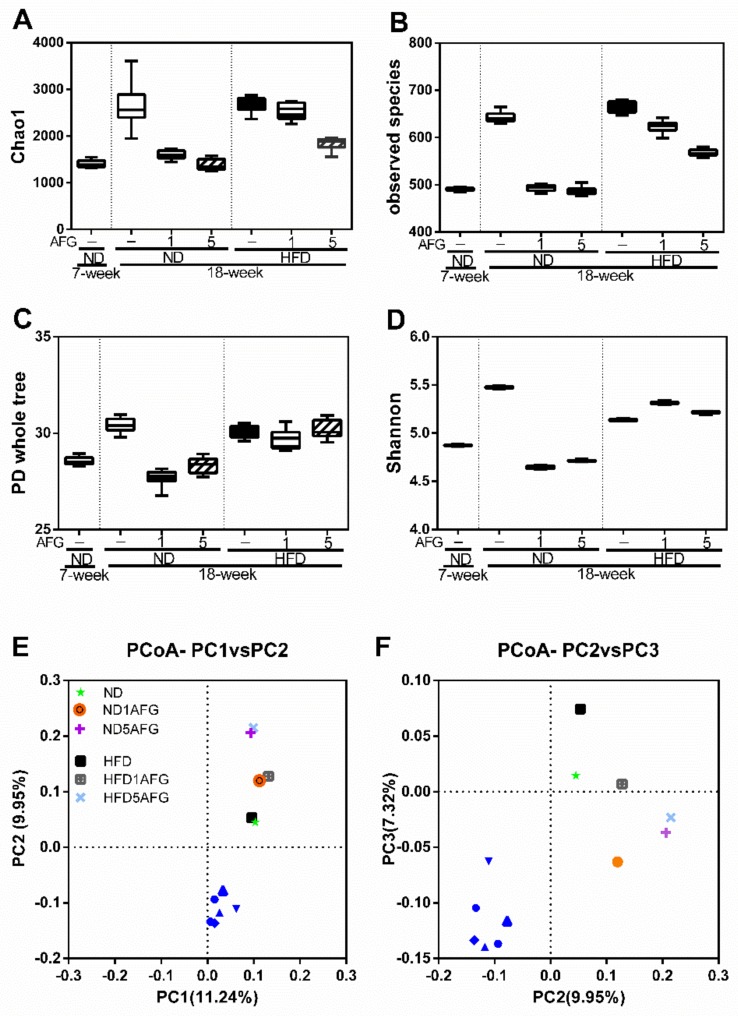
The diversity of gut microbiome. (**A**–**D**) The taxa richness of the gut microbiome assessed by α-diversity analyses using Chao1 value, observed species index, PD whole tree index, and Shannon index. The data represent the median and range of ten alpha rarefaction values. (**E**–**F**) The species compositions of the gut microbiomes were assessed by β-diversity analyses using principle coordinate analysis (PCoA) of the unweighted UniFrac distance matrices, which is showed in PC1 vs. PC2 and PC2 vs. PC3. Each dot represents the beginning (7-week) or ending point (18-week) of the experiment for eight rarefaction values in each group.

**Figure 4 molecules-25-00682-f004:**
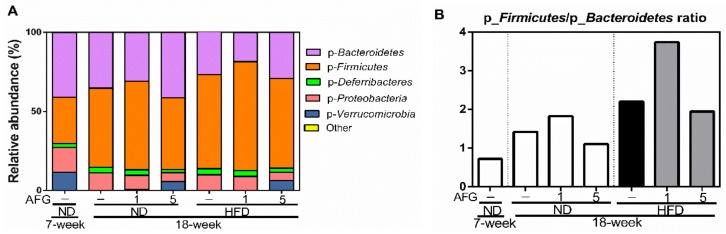
The composition of gut microbiome at phylum level. (**A**) The relative abundance of bacteria at phylum level. (**B**) The ratio of *p*-*Firmicutes* to *p*-*Bacteroidetes* based on their relative abundance.

**Figure 5 molecules-25-00682-f005:**
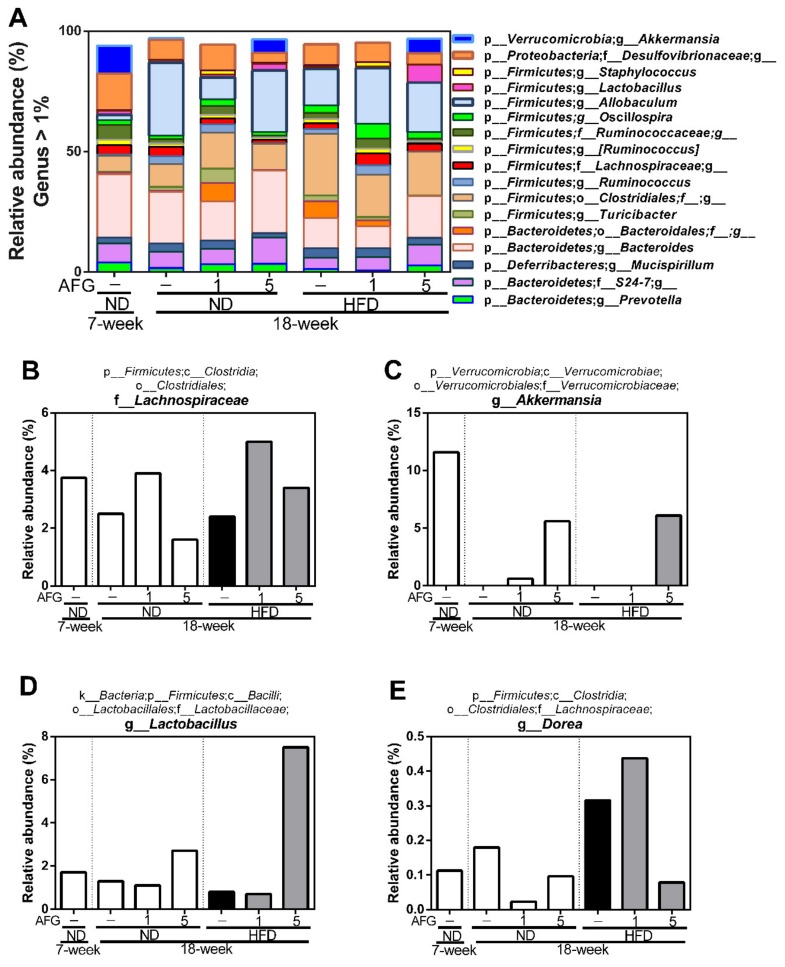
The composition of gut microbiome. (**A**) The relative abundance of more than 1% of bacteria at the genus level. (**B**–**E**) The relative abundance of f-*Lachnospiraceae*, g-*Akkermansia*, g-*Lactobacillus*, g-*Dorea*, respectively. p-, c-, o-, f-, and g- represent phylum, class, order, family, and genus, respectively.

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
