# Peer review of "Modulation of Allicin-Free Garlic on Gut Microbiome"

_molecules, 2020, doi:10.3390/molecules25030682_

Round 1

Reviewer 1 Report

I found experimental data and results very similar (if not identical) to your previous work published in the journal Nutrients.

(Chen K, Xie K, Liu Z, Nakasone Y, Sakao K, Hossain A, Hou DX. Preventive
Effects and Mechanisms of Garlic on Dyslipidemia and Gut Microbiome Dysbiosis.Nutrients. 2019 May 29;11(6). pii: E1225. doi: 10.3390/nu11061225. PubMed PMID:1146458; PubMed Central PMCID: PMC6627858).

Do you have any explanation to give?

Author Response

Response to reviewers’ comments (Reviewer 1)

I found experimental data and results very similar (if not identical) to your previous work published in the journal Nutrients. Do you have any explanation to give?

Response: Our previous paper published in the journal Nutrients reported that the effect of unprocessed whole garlic on gut microbiome and lipid metabolism, and found that allicin, a major antibacterial compound, might contribute these changes. To clarify these, we processed allicin-free garlic (AFG) in this study to investigate the effect of AFG on gut microbiome and lipid metabolism. Therefore, the experiment materials and data are different with that in our previous paper although the experiment style is similar with our previous paper. We also discussed these in Discussion section. Thank you for your understanding.

Reviewer 2 Report

The paper is well written and conclusion as backed up by the experiments.    There are a few small concerns:   The abstract is quite long in the current format, while technically accurate it needs to be more focused and too the point. I would expect it to be half the size it is right now. There are also several grammar errors here.   'Sulfenic acid undergoes self-condensation resulting in allicin', it would be nice to have a figure showing this to build the picture for the reader. It would also help to build a wider context to understand the paper and the significance of this work and it's place within the wider research field.    figure 1A and figure 5A are difficult to read and need to be improved - a higher resolution image needs to be provided   page 4 line 107, page 7 line 166 highlight a broader trend of formatting issue, many spaces missing and there are some typos present

Author Response

Response to reviewers’ comments (Reviewer 2)

The abstract is quite long in the current format, while technically accurate it needs to be more focused and too the point. I would expect it to be half the size it is right now.

Response: Yes, we have shortened our abstract and gave more concise abstract in revised manuscript.

Sulfenic acid undergoes self-condensation resulting in allicin', it would be nice to have a figure showing this to build the picture for the reader. It would also help to build a wider context to understand the paper and the significance of this work and it's place within the wider research field.

Response: We have added a figure (Appendix Figure 1) from the references [12], which explains the biotransformation processes of OSCs in garlic. The author of this references paper [12] from our laboratory, which will not involve copyright issues.

figure 1A and figure 5A are difficult to read and need to be improved - a higher resolution image needs to be provided.

Response: Yes, we have adjusted the image and replace it with a higher resolution image.

page 4 line 107, page 7 line 166 highlight a broader trend of formatting issue, many spaces missing and there are some typos present.

Response: Thank you for your careful reviewing. We have corrected the error.

Original (line 107):Moreover, the ratio of p-Firmicutes to p-Bacteroidetes was increased by aging from 7 weeks to 18 weeks in ND group, and attenuated by 5%AFG supplementation.

Modified:Moreover, the ratio of p-Firmicutes to p-Bacteroidetes in the ND group was increased with aging from 7 weeks to 18 weeks, and decreased by 5%AFG supplementation.

Original (line 166):Centrifuge the garlic homogenate at 5000rpm for 30 min with 4 cycles to remove impurities and extract the supernatant. After centrifugation, the garlic extract was freeze-dried to a garlic powder, which was dissolved in a ratio water of 1: 2. After resting overnight, centrifuge under the same conditions as before, remove the supernatant, add anhydrous ethanol into the precipitate, and leave it overnight after being stirred slightly. Until the precipitate was confirmed to be non-viscous, remove the supernatant. And the precipitate was pulverized and dried by hot air drying at 60 ℃ for 40 min, and then stored at -2℃ out of light.

Modified:After removing water, heated garlic was homogenized and then centrifuged 4 times at 5000rpm for 30 minutes. The supernatant was collected and freeze-dried to garlic powder, which was further washed with ethanol and dried again at 60 ℃ for 40 min to obtain AFG powder.

Round 2

Reviewer 1 Report

The authors responded to my concerns and the paper has been improved